

# Epidemic monitoring in real-time based on dynamic grid search and Monte Carlo numerical simulation algorithm

Xin Chen[1], Huijun Ning[1], Liuwang Guo[2], Dongming Diao[1], Xinru Zhou[3] and Xiaoliang Zhang[1]

[1] College of Civil Architecture, Henan University of Science and Technology, Luoyang, China
[2] School of Mathematics and Statistics, Henan University of Science and Technology, Luoyang, China
[3] School of Information Management and Engineering, Shanghai University of Finance and Economics, Shanghai, China

## ABSTRACT

Building upon the foundational principles of the grid search algorithm and Monte Carlo numerical simulation, this article introduces an innovative epidemic monitoring and prevention plan. The plan offers the capability to accurately identify the sources of infectious diseases and predict the final scale and duration of the epidemic. The proposed plan is implemented in schools and society, utilizing computer simulation analysis. Through this analysis, the plan enables precise localization of infection sources for various demographic groups, with an error rate of less than 3%. Additionally, the plan allows for the estimation of the epidemic cycle duration, which typically spans around 14 days. Notably, higher population density enhances fault tolerance and prediction accuracy, resulting in smaller errors and more reliable simulation outcomes. Overall, this study provides highly valuable theoretical guidance for effective epidemic prevention and control efforts.

## INTRODUCTION

Epidemic diseases have always been a major concern for public health, impacting both human well-being and economic progress. With the rise of globalization and population movements, the frequency and scale of epidemic outbreaks are increasing rapidly. As a result, researchers are dedicated to studying the transmission patterns and strategies for controlling epidemic diseases, aiming to identify and contain outbreaks at an early stage, thereby protecting public health and ensuring social stability. In recent times, mathematical models and computer simulations have emerged as the primary research approaches in the field of epidemic disease research. These methods allow for the analysis and prediction of virus transmission patterns and underlying factors, serving as a scientific foundation for decision-making and the implementation of preventive measures by governments and the general public. Moreover, advancements in gene sequencing and big data analysis technologies have played a crucial role in tracing the origins and transmission routes of

Corresponding author
Huijun Ning, ninghui-jun@haust.edu.cn

epidemic diseases. This enables accurate tracking of viruses and provides more precise scientific support for effective epidemic prevention and control.

Currently, genome sequencing technology is widely employed by researchers worldwide to trace the origins of epidemic diseases. Notably, *He et al. (2022)* utilized the Filmarray pathogen detection system and Norovirus fluorescent polymerase chain reaction detection to screen and isolate bacteria according to the diagnostic criteria for infectious diarrhea (WS 271-2007). Subsequently, they amplified, sequenced, and compared the polymerase region and capsid region sequences of norovirus-positive nucleic acid. Homology analysis was conducted using Mega software. Similarly, *Zhang & Wang (2007)* analyzed the nucleic acid sequences of SARS-CoV and SARS-like coronaviruses (SARS-CoV) from GenBank. They constructed a systematic phylogenetic tree of SARS-CoV and SARS-like coronaviruses, using the Breda virus as the outgroup, and performed single nucleotide variation analysis. Additionally, *Takenouchi et al. (2021)* evaluated the effectiveness of using whole-genome sequencing to identify the source of infection in hospitalized patients with no apparent history of exposure to infection. In conclusion, current epidemic detection technologies accurately diagnose infected individuals and provide a foundation for subsequent mathematical modeling research.

Researchers worldwide have extensively utilized the susceptible-exposed-infectious-removed (SEIR) model and its extensions to predict virus infections based on current epidemic detection techniques. *Cai, Jia & Wang (2020)* applied the SEIR model to forecast the trajectory of the Wuhan epidemic in 2020, estimating its cessation by the end of April. They also proposed varying levels of government interventions for infected individuals at different time points. In 2022, *Dong, Song & Meng (2022)* employed the SEIR-ARIMA hybrid model to predict and analyze the epidemic's progression in different periods and regions, yielding highly accurate results that closely matched the actual situation. *Yarsky (2021)* developed an SEIR model to forecast the spread of the 2021 coronavirus (SARS-CoV-2) in the United States. Genetic algorithms were employed to determine the parameters required for the SEIR model, and reopening strategies and hospital resource utilization were considered to predict the expansion of the epidemic within the population. *Annas et al. (2020)* established an SEIR model incorporating vaccination and isolation factors as model parameters. Using the generation matrix method, they obtained the basic reproduction number and assessed the global stability of the infectious disease distribution model. However, most domestic and international monitoring and prevention plans for epidemics adopt a macro perspective, overlooking various objective factors that impact the scale of epidemic transmission. Existing epidemic prediction plans lack dynamic adjustment capabilities, struggle to accurately simulate real-time epidemic scales, and often fail to pinpoint infection sources precisely, resulting in limited practical value for epidemic prevention and control efforts.

Addressing the aforementioned challenges, this article presents a novel epidemic monitoring and prevention plan that utilizes the dynamic grid search (*Chen, Huang & Liu, 2020*; *Jin, 2020*; *Kong, Ye & Xu, 2007*; *Liao, 2007*; *Zhu & Li, 2016*) and Monte Carlo numerical simulation algorithm (*Dong, Gu & Yang, 2003*; *Yang & Yao, 2022*; *Zhang, 2022*). This plan offers the capability to accurately simulate the real-time scale of infectious

diseases and precisely identify the exact location of infection sources. The objective is to provide more compelling theoretical guidance for the prevention and control of infectious diseases.

## MATERIALS & METHODS

### Establishment of the model
*Monte Carlo algorithm*

The Monte Carlo algorithm is a statistical simulation method that relies on the principles of probability and statistics. It is a numerical calculation approach that associates a given problem with a specific probability model. Through extensive sampling or statistical simulation using a large set of random or pseudo-random numbers, the algorithm obtains an approximate solution for the problem at hand (*Shao, 2012*; *Zhu, 2014*).

The spread of infectious diseases is a stochastic process, where the transmission occurs as a signal from an information source to multiple receiving sources. These receiving sources act as new information sources, potentially transmitting the disease further. However, the decision of whether and how these receiving sources propagate the information again involves probability events. In handling such probability events, Monte Carlo simulation proves to be highly advantageous (*Chen et al., 2022*).

(1) A probabilistic process can be described as a series of events or actions influenced by random factors or uncertain outcomes. In the context of the Monte Carlo algorithm, it involves analyzing and simulating this probabilistic process in a randomized manner. For problems that inherently possess randomness, the algorithm accurately captures and models the probabilistic nature of the process. However, for deterministic problems that lack inherent randomness, a constructed artificial probability process is necessary. In this case, certain parameters of the artificial probability process are designed to represent the solutions to the desired problem. Essentially, a problem that lacks inherent randomness is transformed into a problem with probabilistic properties through the construction of an artificial probability process.

(2) To implement sampling from a known probability distribution, one of the fundamental techniques used in the Monte Carlo method, random variables with specific probability distributions are generated based on the constructed probability model. This is the reason why the Monte Carlo method is often referred to as random sampling. Among the various probability distributions, the uniform distribution on the interval (0, 1) holds significant importance. Random numbers obtained from this distribution serve as subsamples of the uniform distribution and can be generated using mathematical recurrence formulas on a computer. Different methods exist for random sampling from known distributions, and unlike uniform sampling from (0, 1), these methods rely on generating random sequences, which essentially means generating random numbers. Thus, the generation of random numbers serves as the fundamental tool for Monte Carlo simulations.

### Dynamic grid algorithm

In this article, an epidemic real-time monitoring plan is proposed, considering the characteristics of virus infection. The plan involves dividing the population based on different living areas, considering the spread of the epidemic within each designated area (*Su et al., 2020*).

In the event of an undetected carrier of the virus emerging within the population, a uniform and random sampling process is implemented across all partitioned areas to identify infected patients. Currently, domestic nucleic acid testing technology is well-developed, with an average turnaround time of six hours for test results. If positive patients are detected in the test results obtained on the same day, the specific block where these positive patients are located, as well as its surrounding area, will be subject to a lockdown measure (*Zhang et al., 2021*; *Zhong et al., 2020*).

To mitigate the risk of infection spread, as the surrounding blocks have close proximity to the area where positive patients are identified, it is crucial to implement isolation measures simultaneously in both the block with positive patients and the neighboring blocks. Additionally, on the second day, nucleic acid testing is conducted on individuals residing in the surrounding blocks to monitor and detect potential infections among the population in those areas. This proactive approach aims to prevent the further transmission of the epidemic beyond the initial block and promptly identify and respond to any potential outbreaks in the surrounding vicinity.

In the scenario where not all infected individuals can be detected through nucleic acid testing on the first day, it is important to consider the possibility of undetected infected individuals continuing to transmit the virus to people in their vicinity. To simplify the problem, it is assumed that an undetected infected person can only infect individuals in the eight adjacent partitioned areas. In subsequent days, the testing strategy follows a fixed-point approach, where the same uniform random sampling test points used on the first day are utilized. This means that the testing locations remain consistent throughout the monitoring period. The aforementioned steps are repeated for long-term testing until there are no confirmed cases for 14 consecutive days. This indicates that the epidemic in the region is under control. Different symbols or representations are used to denote detection points, infection points, and quarantine areas, as illustrated in Fig. 1.

For the convenience of analysis, the regional population is divided into an $n \times n$ grid consisting of square areas with n rows and n columns. Each block area within the grid is represented by a two-dimensional vector (x, y), indicating its position in the x row and y column. To optimize the utilization of medical resources, a random sampling test is conducted on a subset of these $n^2$ blocks. The size of the sampling area is determined as $[20\%, n^2]$, where [x] denotes the largest integer less than or equal to x. Subsequently, the specific block area to be tested is identified, and its coordinates are recorded, commonly referred as $(x_1, y_1), (x_2, y_2), \dots, (x_{[20\%n^2]}, y_{[20\%n^2]})$. Equation (1) is defined to record the position of the detection point on the first day.

$$S_1 = \left\{ (x_1, y_1), (x_2, y_2), \dots, (x_{[20\%n^2]}, y_{[20\%n^2]}) \right\}. \tag{1}$$

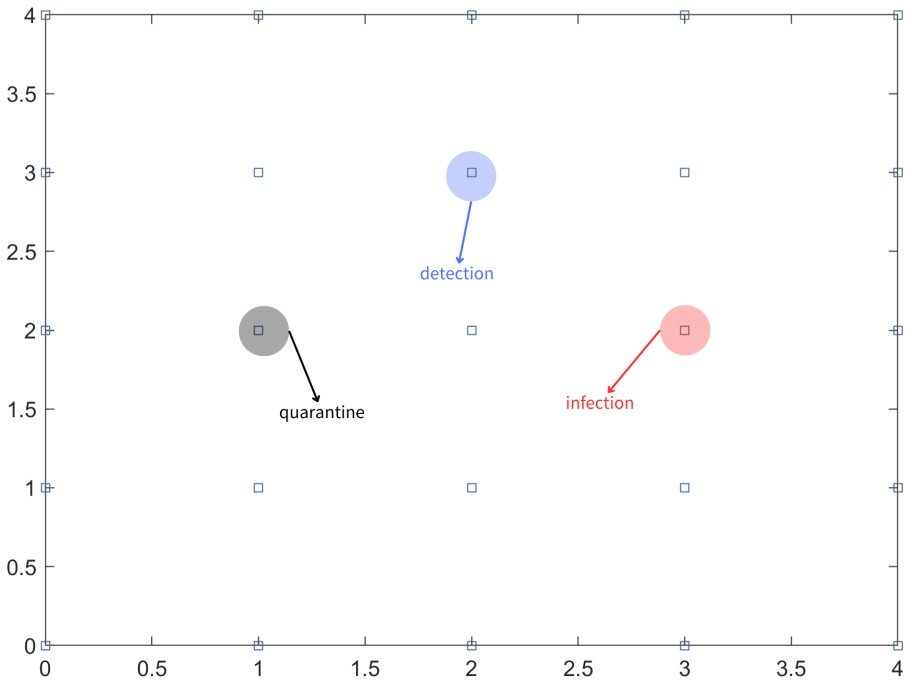

**Figure 1** Block diagram of the area, infection points, detection points and quarantine.

Equation (2) displays the location coordinates of the surrounding block areas if the infected person is detected on day T, and the coordinates of this detection point are known.

$$
\begin{cases}
(x_i - 1, y_i) \\
(x_i + 1, y_i) \\
(x_i, y_i - 1) \\
(x_i, y_i + 1) \\
(x_i - 1, y_i + 1) \\
(x_i + 1, y_i - 1) \\
(x_i - 1, y_i - 1) \\
(x_i + 1, y_i + 1)
\end{cases}
\tag{2}
$$

Equation (3) represents the location set of the isolated area on day T.

$$
\begin{aligned}
F_t = \big\{ &(x_i - 1, y_j), (x_i + 1, y_j), \\
&(x_i + 1, y_j - 1), (x_i, y_j - 1), \\
&(x_i + 1, y_j - 1), (x_i, y_j), \\
&(x_i - 1, y_j + 1), (x_i, y_j + 1), \\
&(x_i + 1, y_j + 1) \big\}
\end{aligned}
\tag{3}
$$

Equation (4) demonstrates the set relationship between the location set $A_t$, which represents the block area where the infectious virus is detected on day T.

$$
A_t \subset S_t \cup F_t
$$
$$
S_{t+1} = (S_t \cup F_t) - A_t
\tag{4}
$$

Based on the analysis of the set relationship, it is assumed that the epidemic is under control on day T. This means that on the day (t-14), all infected block areas are detected, and the adjacent block areas are isolated. Consequently, the sets An, Fn, and Sn are all stable. The mathematical relationship is expressed as $A_n = A_{t-14}, F_n = F_{t-14}, S_n = S_{t-14}$ $(n \geq t)$. Hence, the determination of the steady state can serve as a means to assess and control the epidemic situation in the region.

The iteration mode of variables in each dynamic grid search is shown in Equation (5).

$$\begin{cases} A_n = A_{n-1} + \text{new}_{A_n} \\ F_n = F_{n-1} + \text{new}_{F_n} \\ S_n = rand + F_n \\ d_n = \sum_{j=1}^{\text{length}(F_n)} \sum_{k=1}^{\text{length}(F_n)} \left\{ \begin{array}{l} [F_n(j,1) - F_n(k,1)]^2 + \\ [F_n(j,2) - F_n(k,2)]^2 \end{array} \right\} \\ source = F_n[\min(d_n)] \end{cases} \tag{5}$$

Here, $\text{new}_{A_n}$ represents the newly added infection points, $\text{new}_{F_n}$ represents the newly added isolation points on the nth day, and $length(F_n)$ represents the $F_n$ number of elements.

Once the spread of the epidemic is contained in this phase $d_n$, Equation (5) calculates the sum of Euclidean distances between the nth infection point and all infection points. The smallest $d_n$ corresponding corner marker N is then selected as an approximation for the coordinates of the initial infected person $F_n[(x_n, y_n)]$.

# RESULTS

## Establishment of epidemic monitoring program model

In this article, we devise a comprehensive epidemic monitoring strategy for both universities (with a total population of 50,000) and cities (with a total population of 6 million). Our objective is to efficiently track the initial infected individual while minimizing the number of people subjected to testing, all while accurately forecasting the duration of the outbreak's spread.

Given the close contact among students within the same major in universities, this study proposes a divisional approach based on academic disciplines. A combined collection of nucleic acid tests is performed for all staff within a specific major, which is referred to as a mixed sample test (*Yan et al., 2019*). Considering the relatively small number of individuals within any given major, comprehensive nucleic acid testing is performed for students, which is termed a single sample test. In the event of a positive case within an area where students of the same major are situated after division, individuals within that area will be quarantined, while other areas surrounding the major will be isolated. Subsequently, if positive cases are identified in other areas, mixed sample tests will be conducted the following day, repeating the aforementioned process. The model developed in this research enables dynamic tracking of the epidemic's spread, estimation of the duration of the current outbreak, and identification of the first infected person. As the outbreak tends to propagate from the infection center to its surroundings, this article defines an area where students of the same major are located as an infection point, approximating the collective centroid

of all infection points as the initial patient's presumed location within the major. Each student within the major is then considered an independent infection point, and the first infected person's coordinates are retraced based on area monitoring methods. Due to the asymmetry of infection points, the location of the first patient is determined by calculating the minimum sum of Euclidean distances from one infection point to all others.

Given that cities are predominantly composed of communities, wherein individuals within the same community frequently interact, this study opts to examine cities from a community perspective. Drawing upon the monitoring strategies employed by universities, it becomes feasible to forecast the duration of the ongoing wave of the epidemic and ascertain the timing of the initial infection.

### Model assumptions

To simplify and streamline the problem-solving process, the model is built upon the following assumptions:

(1)  There are 100 majors in a college, and the number of people in each major is 500.
(2)  There are 2,000 communities in the city, and the number of people in each community is 3,000.
(3)  The nucleic acid test results are highly accurate, with no occurrences of false negatives or false positives.
(4)  The nucleic acid test results are conducted and released on the same day.
(5)  There is a 100% infection rate from infected points to surrounding points.
(6)  The quarantined population is immune to both infection and transmission.
(7)  All individuals will willingly adhere to the measures of the monitoring program and actively cooperate.
(8)  There are no sudden increases in the transmission of the epidemic, aligning with the principle of contiguous contagion.
(9)  There are eight neighboring individuals to an infected point.
(10)  Each cycle of epidemic transmission can be contained within a specific timeframe.
(11)  The medical reserve resources are sufficient.

### Establishment and solution of epidemic monitoring model in colleges and universities

For colleges and universities with a population of 50,000, a division of the student body into major-based groups is implemented due to increased interactions among individuals of the same major. Assuming there are 100 majors in the school, with each major consisting of 500 individuals, the major areas are simplified as $10 \times 10$ square blocks. Each major is considered an infection point with specific horizontal and vertical coordinates. Within this model, an infected major can only transmit the infection to the surrounding eight adjacent majors. Additionally, a fixed-point nucleic acid test is conducted uniformly and randomly on 20% of the $10 \times 10$ block area (referred to as mixed sample tests), while each individual within a specific major undergoes a single test (known as single sample tests) to identify any infected individuals. By employing the established model, we can approximate the location of the first infected person within a specific major and determine the duration in days for the current round of the epidemic to spread among different major areas.
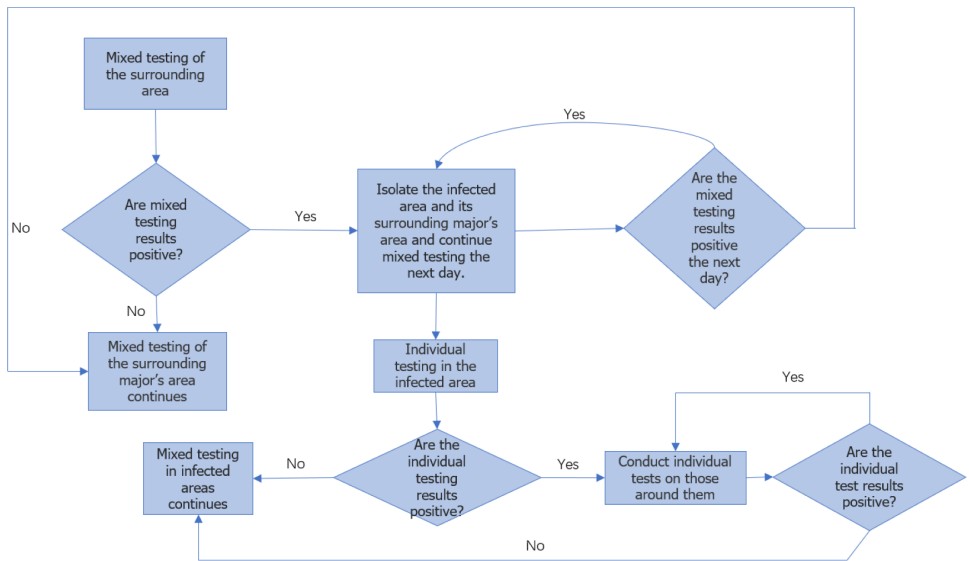

**Figure 2   Flow chart of epidemic monitoring in colleges.**

Figure 2 illustrates the flowchart for college monitoring. Once the approximate location of the major of the first infected person is determined, the individuals within that major are further divided into 24×24 areas for analysis, considering each person as an independent infection point. By employing a similar approach as professional traceability, we approximate the coordinates of the first infected person within the major and predict the duration of epidemic transmission among individual members in this particular round. Once the ongoing round of the epidemic in colleges and universities is effectively managed, we can gather mixed sample data on the current round of transmission detection among majors, as well as single sample test data on transmission detection between individuals. This data enables us to obtain the approximate coordinates of the first infected person's major and the approximate coordinates of the first infected individual. Moreover, it is important to note that the virus spreads synchronously between different majors and within the major. Therefore, by taking the maximum value of the transmission time between majors and individuals in this round of the epidemic, we can approximate the overall duration of this particular round.

In order to validate the model's accuracy, this article assumes the availability of a known infection point. The model proposed in this study is then employed to address this scenario. The objective is to trace the approximate geographical coordinates of both the major and individual origins of the initial infected person. An error analysis is conducted by comparing these coordinates with the provided hypothetical infection point. Additionally, by considering the maximum time taken to trace the approximate coordinates of the major and the individual back to the source of the first infection, we can approximate the duration of this current epidemic cycle. This process can be described as a dynamic grid search.

During each iteration of the dynamic grid search, a random location coordinate is assigned to the initial infected person, denoted as the "Center." The initial nucleic acid test

point, referred to as the "Rand," is uniformly and randomly determined within the global area. By utilizing the model above, we can predict the approximate location coordinate of the first infected person, denoted as the "Source," after the conclusion of this epidemic cycle.

This article utilizes the concept of the Monte Carlo algorithm to address the limitations and unpredictability associated with a single solution. By iteratively running the dynamic grid solution process multiple times, the average value of multiple predictions is calculated and considered as the final prediction outcome. This approach aims to overcome contingencies and enhance result stability.

Initially, the infection point is positioned at Center (5, 5) of the major area. Following five simulations, the average duration for epidemic control across different major areas is determined to be 16.5 days, while the estimated number of mixed mining reaches 59.14. The last traceability coordinate calculated is Source (5, 5.14). The relative error for the horizontal and vertical coordinates of the initial simulation point, Center (5, 5), is reported to be (0%, 2.8571%).

For the position of the initial infected person in the area block Center (5, 5), we take Center (12, 12). After running 100 simulations, the average duration for epidemic control within the profession is found to be 18.01 days, and the estimated number of single sample tests is 446.64. The final traceability result indicates the source as (12, 12.09). The relative error for the horizontal and vertical coordinates is reported to be (0%, 0.75%). These results demonstrate that the relative error of the major's horizontal and vertical coordinates obtained through source tracing does not exceed 3%, while the relative error for the initial infected person's coordinates derived from the source tracing remains under 1%. Consequently, the model's accuracy is validated.

In summary, the final solution result reveals that the duration of the epidemic in colleges and universities for this round is 18.01 days. On average, there have been 59.14 instances of mixed mining and 30016.64 instances of single mining. Figure 3 presents a visual representation of two simulation results during the dynamic grid search process.

### Establishment and solution of the urban epidemic monitoring model

In a city with a population of 6 million, we make the assumption that there are 2,000 neighborhoods, with each neighborhood consisting of approximately 3,000 people. To facilitate analysis and simulation, we further divide the community into $46 \times 46$ areas. Each of these areas is then subdivided into smaller $56 \times 56$ regions for consideration. According to the epidemic transmission rule, infected individuals within a given area can only transmit the infection to people residing in the surrounding areas.

In the first step, we randomly select 20% of the $46 \times 46$ block areas for fixed-point nucleic acid testing, which is referred to as mixed sample tests. Then we conduct single sample tests on all individuals within a selected community area to identify any infected individuals within the community. By employing the established model, we can approximate the location of the community where the first infected person is situated and determine the duration of the epidemic for the respective community areas.

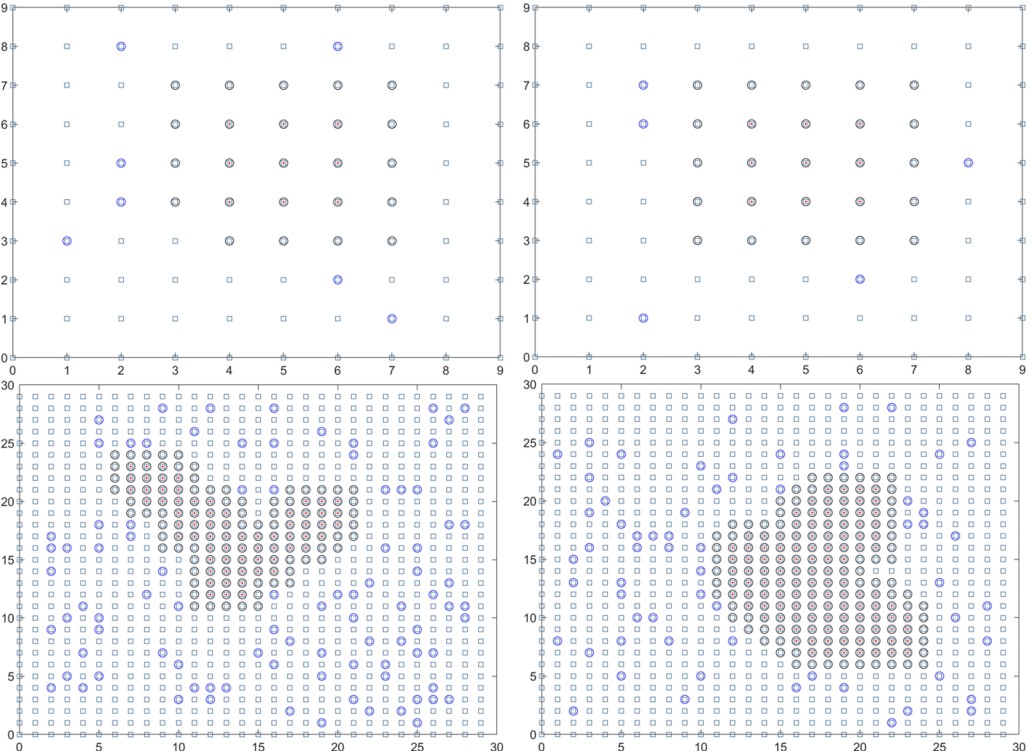

**Figure 3** **Propagation state diagram when part of the simulation is stable.** (A) The first simulation to simulate the propagation state diagram between stable majors. (B) The second simulation simulates the propagation state diagram between stable majors. (C) The third simulation simulates the internal propagation state diagram of the stable major. (D) The fourth simulation of the internal propagation state diagram of the stable major.

Once we have obtained the approximate location, the individuals within that community are further divided into 56×56 areas for analysis. In this case, each person is treated as an independent infection point. Similar to community tracing, we employ the same methodology to estimate the approximate coordinates of the first infected individual and predict the duration of epidemic transmission among the individual members. The monitoring process is illustrated in Fig. 2, which depicts the flowchart of the monitoring procedure. Once the current round of urban epidemic is brought under control, we can gather mixed sample test data for detecting transmission between communities and single sample test data for detecting transmission between individuals. Additionally, we can obtain the approximate coordinates of the community where the first infected person is located and the approximate coordinates of the individual who is the first to be infected. It is important to note that the spread of the virus occurs synchronously between community areas and within communities. Therefore, by considering the maximum value of the transmission time between communities and individuals during this round of the epidemic, we can approximate the overall duration of this round of the epidemic.

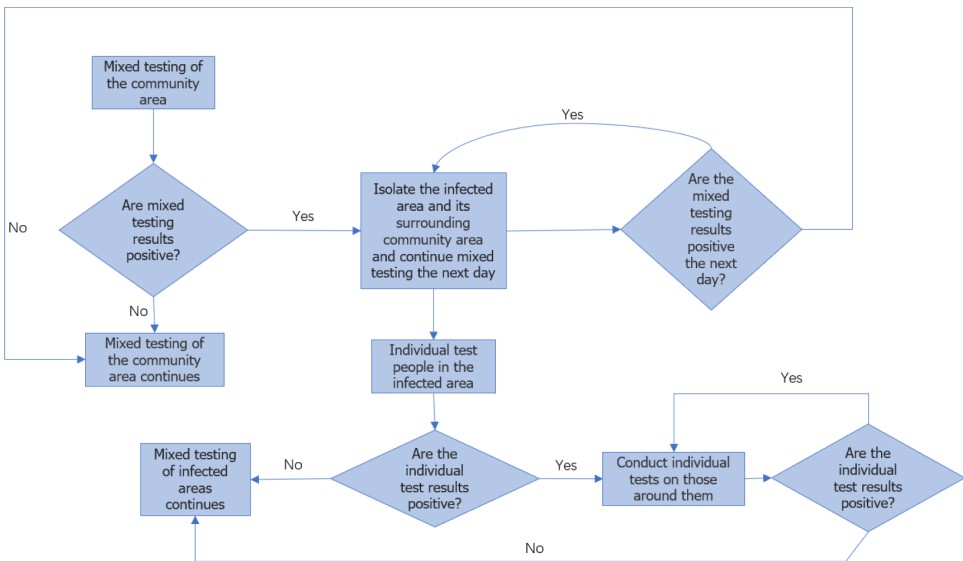

**Figure 4** Flow chart of urban epidemic monitoring.

The specific solution process is the same as the university model. The flow chart can be seen in Fig. 4.

To overcome the limitations of a single solution, this article incorporates the Monte Carlo algorithm. The dynamic grid solution process is iterated multiple times, utilizing the concept of a loop. By doing so, multiple predictions are generated. To enhance result stability and reduce contingency, the final prediction result is obtained by calculating the average value of these multiple predictions.

The initial infection point is located at Center (23, 23) of the community block area. Through 1,000 simulations, the average duration of epidemic control between community areas is determined to be approximately 18.488 days. The estimated number of mixed sample tests is calculated to be 1727.324. The last traceability coordinate obtained is Source (22.9885, 22.9925). The relative error for the horizontal and vertical coordinates of the initial simulation point is reported to be (0.05%, 0.032609%).

In the case of the initial infection within the population of the community block, the center point is adjusted to (28, 28) for analysis. After conducting 1,000 simulations, the average duration for epidemic control within the community is determined to be approximately 18.285 days. The expected number of single sample tests is calculated as 2419.455. The final traceability result indicates the source as (28.015, 28.011), with a relative error of (0.053571%, 0.039286%) for the horizontal and vertical coordinates. These results demonstrate that the relative error for the horizontal and vertical coordinates of the community where the first infected person is located does not exceed 0.05%, while the relative error for the horizontal and vertical coordinates of the first infected individual obtained from the source tracing remains below 0.06%. Hence, the accuracy of the model is verified. In summary, the final solution for this round of the epidemic in the city is as follows: the average duration is determined to be 18.488 days. Throughout the epidemic,

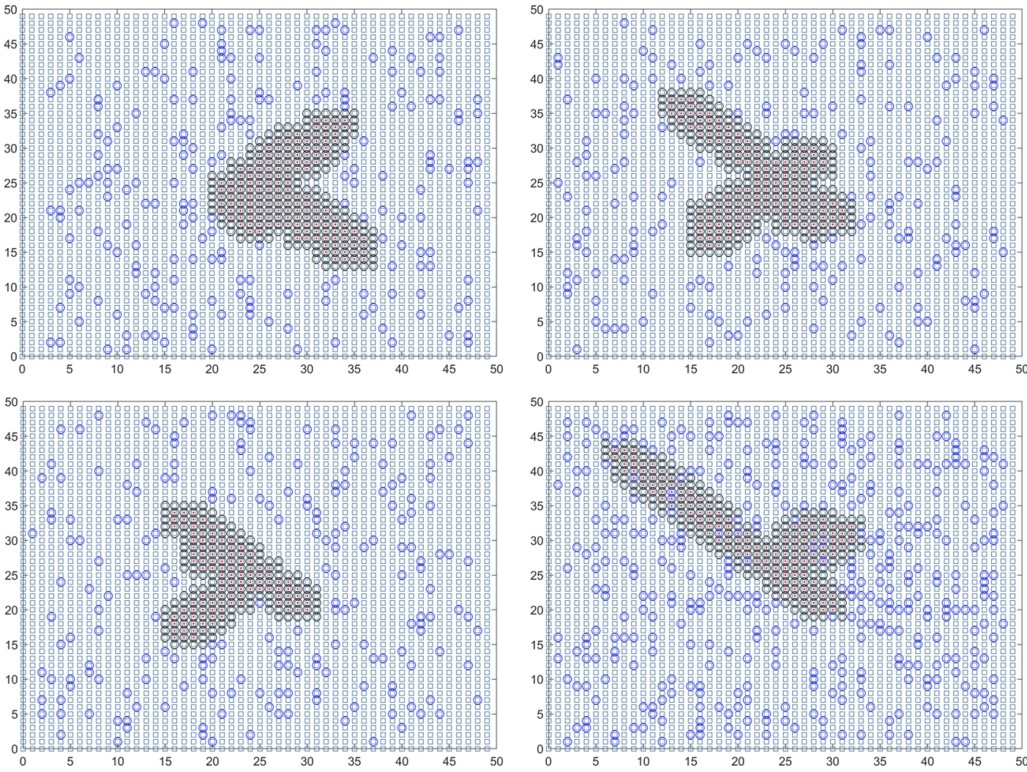

**Figure 5  Propagation state diagram when the simulation of the urban part is stable.** (A) The first simulation simulates the state diagram of the spread between stable communities. (B) The second simulation simulates the propagation state diagram between stable communities. (C) The third simulation simulates the internal communication state diagram of the stable community. (D) The fourth simulation simulates the internal transmission state diagram of the stable community.

there has been an average of 1727.324 instances of mixed mining and 5.18 million instances of single mining. Figure 5 provides a visual representation of two simulation results during the dynamic grid search process.

## DISCUSSION

Employing advanced techniques in epidemic detection enables the differentiation of virus carriers within a population. Consequently, the newly devised plan for monitoring and preventing epidemics simulates the transmission of the disease starting from the initial patient under monitoring.

Under the assumption of no spatial–temporal cross-infection among patients and the absence of external factors during the transmission process, a specific proportion of the epidemic is sampled for testing purposes. Detected patients and their close contacts are subjected to isolation and observation policies, during which both the patients and the surrounding population are considered incapable of infecting others. Undetected patients are assumed to have a 100% infection rate and can continue spreading the disease to the surrounding population. The new monitoring and prevention plan concludes the

simulation and provides the scale and duration of the epidemic when there are no new infection cases reported for fourteen consecutive days.

In further research, social investigations can be used to establish spatiotemporal connections between patients and accurately model the transmission of viruses across different locations and timeframes. Furthermore, monitoring and prevention plans can be enhanced by integrating objective factors such as population immunity, infection source variability, and environmental suitability for transmission. By considering these factors, the simulation results can be made more realistic and valuable in informing effective strategies for combating the spread of diseases.

## CONCLUSIONS

This study introduced a novel approach for real-time monitoring and tracking of infectious disease spread by dividing the population into modules. Specifically, a university with 50,000 individuals was divided into two modules: majors and students, while a city with six million residents was divided into communities and residents. By employing the dynamic grid search method, the researchers conducted real-time simulations to track the source of infection and determine the location coordinates of the first infected individual. Multiple simulations were carried out using the Monte Carlo algorithm to eliminate chances and biases, and the average values were considered as the final predictions. The findings are as follows:

1. For colleges and universities of 50,000 people, the average duration of the epidemic infectious disease is 18.01 days. On average, 59.14 mixed sample tests and 30,016.64 single sample tests were performed.

2. For cities of 6 million people, the average duration of the epidemic infectious disease is 18.488 days. On average, 1,727.324 mixed sample tests and 5.18 million single sample tests were conducted.

3. This model successfully determined the coordinates of the infection source. The error between the traced coordinates and the actual coordinates within the university was less than 3%, while the error within the city was less than 0.06%. Increasing the number of Monte Carlo simulations improved the prediction accuracy. These results indicate that the established model is feasible and accurate in identifying the source of infection, predicting real-time virus outbreak durations, and estimating resource requirements.

To sum up, this study introduced a novel epidemic monitoring and prevention plan that effectively simulated real-time epidemic scale and accurately identified the source of infection, enabling dynamic monitoring of epidemics. In the future, the spatiotemporal capabilities of this plan can be expanded through social surveys to dynamically simulate the presence of virus carriers in different locations at different times, thereby enhancing the realism of monitoring results. Additionally, the simulation and monitoring process can incorporate objective factors influencing epidemic spread, such as the immunogenicity of the transmission group, instability of the infection source, and environmental suitability for transmission. These future advancements will enhance the global robustness of the epidemic monitoring and prevention model and yield more precise results.

### Funding
The authors received no funding for this work.

### Competing Interests
The authors declare there are no competing interests.

### Author Contributions
- Xin Chen conceived and designed the experiments, performed the experiments, analyzed the data, performed the computation work, prepared figures and/or tables, authored or reviewed drafts of the article, and approved the final draft.
- Huijun Ning conceived and designed the experiments, performed the experiments, analyzed the data, performed the computation work, prepared figures and/or tables, authored or reviewed drafts of the article, and approved the final draft.
- Liuwang Guo conceived and designed the experiments, performed the experiments, analyzed the data, performed the computation work, prepared figures and/or tables, authored or reviewed drafts of the article, and approved the final draft.
- Dongming Diao conceived and designed the experiments, performed the experiments, analyzed the data, performed the computation work, prepared figures and/or tables, authored or reviewed drafts of the article, and approved the final draft.
- Xinru Zhou conceived and designed the experiments, performed the experiments, analyzed the data, performed the computation work, prepared figures and/or tables, authored or reviewed drafts of the article, and approved the final draft.
- Xiaoliang Zhang conceived and designed the experiments, performed the experiments, analyzed the data, performed the computation work, prepared figures and/or tables, authored or reviewed drafts of the article, and approved the final draft.

### Data Availability
The computer codes are available in the Supplementary Files.

### Supplemental Information
Supplemental information for this article can be found online at http://dx.doi.org/10.7717/peerj-cs.1479#supplemental-information.

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
