# Peer review of "Epidemic monitoring in real-time based on dynamic grid search and Monte Carlo numerical simulation algorithm"

_PeerJ Computer Science, doi:10.7717/peerj-cs.1479_

## Round 0.1 · original submission · Minor Revisions

Based on reviewers' comments the manuscript needs revisions.

Reviewer 1 ·

Basic reporting

- English text needs revision. It is suggested that the author use tools such as Grammarly to check grammatical mistakes.

- The paper's contribution needs to be stated more clearly.

- The Materials & Methods section could benefit from an increase in references to reliable sources.

- Authors should compare their work in detail with related work.

- Please also add future work and research opportunities to the Conclusion.

Experimental design

Need improvement

Validity of the findings
* * *
Additional comments
* * *
Reviewer 2 ·

Basic reporting

Dear author, please consider the following comments to improve your paper
1- The abstract of this paper does not express the contribution of the work clearly. The abstract text must include the methods used in a new way. The result and comparison part needs to be improved.
2- I suggest inserting complementary information about your idea in the introduction section and categorizing the whole of the work.
3- Please correct the typography errors.
4- The Section titled Materials and Methods contains a few of the related references.
5- There is a loss of useful information about the workflow of your idea before the Result Section.

Experimental design

no comment

Validity of the findings

no comment

Additional comments

no comment

---

## Round 0.2 · accepted · Accept

Based on the reviewers' comments, the manuscript can be accepted.

Reviewer 1 ·

Basic reporting

The authors have addressed the majority of the key comments raised during the initial review process and the revisions made have improved the quality of the manuscript. I recommend that the paper be accepted for publication.

Experimental design

Ok

Validity of the findings

Ok

Additional comments

No Comment

Reviewer 2 ·

Basic reporting

-

Experimental design

-

Validity of the findings

-